# Workplace bullying, psychological hardiness, and accidents and injuries in nursing: A moderated mediation model

**Stephen T. T. Teo** [1]*, **Diep Nguyen**[1], **Fiona Trevelyan**[2], **Felicity Lamm**[3], **Mark Boocock**[4]

**1** School of Business and Law, Edith Cowan University, Western Australia, Australia, **2** School of Clinical Sciences, Auckland University of Technology, Auckland, New Zealand, **3** The Centre for Occupational Health and Safety Research, Auckland University of Technology, Auckland, New Zealand, **4** Department of Physiotherapy, Auckland University of Technology, Auckland, New Zealand

* s.teo@ecu.edu.au

**Data Availability Statement:** The full data set cannot be shared publicly because formal approval was not granted by the Ethics Committee. However, all relevant data necessary to replicate

## Abstract

Workplace bullying are prevalent among the nursing workforce. Consequences of workplace bullying include psychological stress and workplace accidents and injuries. Psychological hardiness is proposed as a buffer for workplace bullying and psychological stress on workplace accidents and injuries. This study adopted the Affective Events Theory and Conservation of Resources Theory to develop and test a moderated mediated model in two field studies. Study 1 (N = 286, Australian nurses) found support for the direct negative effect of workplace bullying on workplace accidents and injuries with psychological stress acting as the mediator. The mediation findings from Study 1 were replicated in Study 2 (N = 201, New Zealand nurses). In addition, Study 2 supplemented Study 1 by providing empirical support for using psychological hardiness as the buffer for the association between psychological stress and workplace accidents and injuries. This study offers theoretical and empirical insights into the research and practice on psychological hardiness for improving the psychological well-being of employees who faced workplace mistreatments.

## Introduction

Workplace bullying is a typical psychosocial risk factor universally prevalent in most workplaces around the world. The Workplace Bullying Institute's 2017 survey reported approximate 40% of the bullied targets reported suffering adverse health effects and this incidence affected 60.4 million Americans [1]. Fevre and colleagues reported that approximately half of the participants in the United Kingdom experienced some forms of unreasonable treatment at work and 40% reported workplace disrespect to be the most common phenomenon [2]. Other studies showed that health sector employees are one of the most vulnerable population to expose to psychosocial risk at work [3]. As previously reported, 65% of nursing professionals in the USA observed lateral violence among co-workers [4]. These statistics highlighted the severity of workplace bullying on the stress of the nursing workforce.

Despite several definitions of workplace bullying [5], the present study adopts the definition of workplace bullying by Einarsen and colleagues [6] as "harassing, offending, socially

the study's results are within the paper and its Supporting Information files.

**Funding:** Funding was provided by the Auckland University of Technology as a research grant awarded to Teo, Lamm and Boocock. Authors Trevelyan, Lamm, and Boocock are full time employees of the funder (Auckland University of Technology).The funders had no role in study design, data collection and analysis, decision to publish, or preparation of the manuscript.

**Competing interests:** The authors have declared that no competing interests exist.

excluding someone or negatively affecting someone's work tasks." Workplace bullying is a major source of psychosocial stressors [7, 8] and it is associated with workplace injury compensation claims [9].

An outcome of psychosocial risk factors is accidents and injuries [10]. Common forms of workplace accidents and injuries among hospital workers include overexertion, falling slips, trips, and falls, contact with objects or equipment, violence, and an exposure to harmful substances [10]. These injuries lead to employees taking sick leaves from work. However, there is inconclusive of the association of psychological stress with workplace accidents and injuries [10, 11]. Therefore, more attention is needed to enhance the workplace safety of nurses [12].

Scholars have been urged to investigate into workplace bullying in the context of workplace safety [12]. A meta-analytical review by Christian et al. showed most studies examine the impact of safety climate and personality factors [13]. Research showed that psychological hardiness could be treated as an important 'resistance' resource [14] which helps employees effectively cope with stressful situations and/or negative work-related events because of the ability of psychological resilience [15], as explained by the Conservation of Resources (COR) theory [16]. However, very little is currently known about the moderating role of psychological hardiness in assisting nurses to cope with workplace bullying and its consequent outcomes although this factor is a potential moderator of stress [17]. We will take up this challenge by proposing that psychological stress is a mediator and psychological hardiness is a resource which could be used to buffer the influences of workplace bullying on workplace accidents and injuries among nurses (Fig 1).

## Theoretical background and hypotheses

### Workplace bullying and psychological stress

A "good" work environment is associated with better work outcomes such as lower stress and injury rates [18]. Workplace bullying is an example of "unsafe" psychosocial work environment [19]. Studies showed that bullying leads to burnout [20], which adversely affects the physical and mental health of nurses [35]. These symptoms are prevalent in nursing, irrespective of gender, age, race, education levels, or work history [21]. In Australia and New Zealand, nurses also experienced workplace bullying and work harassment [22].

The integration of the affective events theory (AET) [23] and the COR theory [16] creates a potentially useful framework to explain the negative effects of workplace bullying on nurses. Drawing from the AET, we argue that when employees experience workplace bullying that is a negatively affective event, they would react emotionally to it which would affect their subsequent well-being [5]. Consistent with the COR perspective, an exposure to workplace bullying could result in the depletion of personal and job resources leading to poor psychological health state [24]. There has been empirical support for the negative association between workplace bullying and psychological stress, as shown in a sample of 233 hospital and 208 aged care nurses from Australia [25]. When nurses are exposed to workplace bullying, they experienced more burnout as they could not recover psychologically [26].

**H1**: *Workplace bullying has a positive association with psychological stress.*

The resulting psychological stress due to workplace bullying could lead to workplace safety outcomes among nurses. The Institute for Safe Medication Practices reported that 7% of 1,565 nurses were involved in medication errors as a result of experiencing intimidation at work [27]. Instead of asking for help in an environment where bullying is present, nurses muddle through an unclear procedure, use an unfamiliar piece of medical equipment without seeking help, lift heavy, or debilitated patients alone [28]. These actions not only are prone to cause

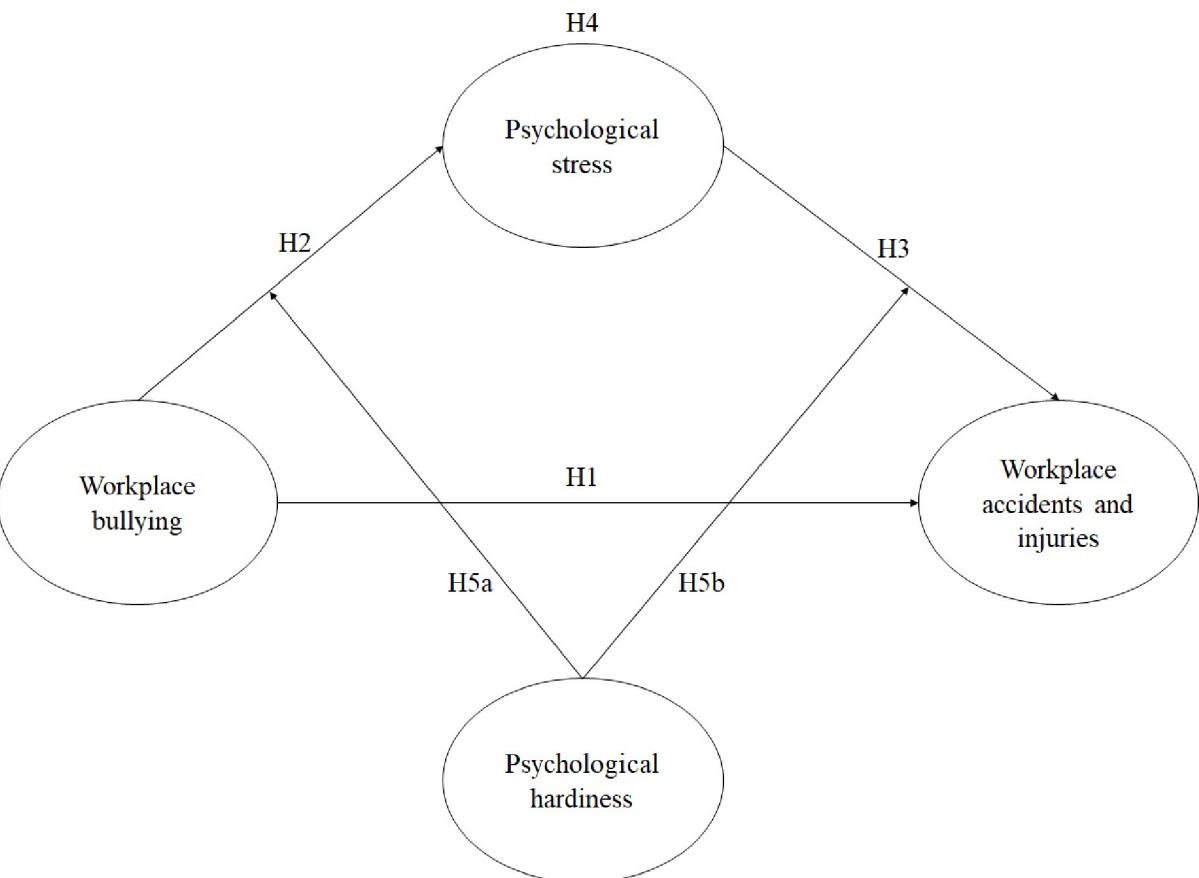

**Fig 1. Proposed moderated mediation model.** *Note*: Control variables: tenure, gender, supervisory role, age and marker variable (social desirability), Study 1: Hypotheses 1 to 4, Study 2: Full model (Hypotheses 1 to 4 and moderation hypotheses).

accidents that compromise patient care and safety, but also can cause injuries to themselves and jeopardize their safety. Salminen et al. provided empirical support between injuries and interpersonal relationship problems as examples of workplace bullying behaviors [11]. Other studies found that workplace bullying among nurses could result in stress-related symptoms such as accidents and errors [29] and negatively affects the quality and safety of patient care [30]. A recent systematic review supported the positive association between workplace bullying and injuries as "unsupportive social relationships" are related to higher levels of employee injury [31].

**H2**: *Workplace bullying has a positive association with workplace accidents and injuries.*

## Psychological stress and workplace accidents and injuries

Workplace bullying has been found to lead to poor mental well-being [5] and emotional exhaustion [9]. AET perspective could be used to explain these relationships [5]. These health problems could cause a loss of concentration and vigilance at work that increases the chance of making mistakes and the likelihood of work accident and injury events, both physical (e.g., needle injuries) and psychological (e.g., violence) [32]. Bullying was perceived to be associated with occupational injuries [33]. There is also evidence supporting the association between bullying and suicidal ideation or attempted suicide, which is the ultimate injury due to extreme psychological stress [34].

**H3**: *Psychological stress has a positive association with workplace accidents and injuries.*

Workplace bullying could also result in sleep disorders and fatigue as well as moderate and high levels of psychological stress that increase the risk of work-related accidents and injuries [35]. To reduce the physiological stress of being bullied, some employees even developed excessive alcohol consumption that is a risk factor for accidents and injuries at work to cope with bullying [36]. Others [37] argued that the association between bullying-related psychological stress and work-related accidents and injuries was caused by nurses' cognitive failures in performing their daily tasks.

**H4**: *Psychological stress mediates the positive association of workplace bullying on accidents and injuries at work.*

## Moderation effects of psychological hardiness

Psychological hardiness has received an increasing interest in high-stress, first respondent occupations such as military [38] and nursing [39]. Hardiness is "a constellation of personality characteristics that function as a resistance resource in the encounter with stressful life events" [40]. A hardy personality comprises of commitment, control, and challenge attitudes that provide a pathway to resilience that facilitates the possibility to turn "stressful circumstances" from adversity into advantage and growth opportunities [14, 41]. Hardy individuals experience stressful work situations in ways that they appraise the potentially stressful situations as less threatening, thus, minimizing their experience of stress [38]. Indeed, research has long recognized that psychological hardiness acts as a protective or buffer factor in coping with work-related stress [42] and a resilience factor against the development of PTSD syndromes [43]. A recent study noted the importance of developing the resilience of nurses in New Zealand in creating a practice environment which reduces workplace bullying [44]. Consistent with these arguments, we argue that psychological hardiness is important in helping nurses cope with workplace bullying.

The COR theory [16] could be used to hypothesize a moderation model where bullying leads to resource depletion among nurses who have resources and where this depletion subsequently reduces health and well-being. The COR theory posits that "people strive to retain, protect and build resources and that what is threatening to them is the potential or actual loss of these valued resources" [16]. Under this theory, workplace bullying is considered as a stressful working condition which brings about employees' experience with a substantial loss or a depletion of both personal and job-related resources [45]. Therefore, they would utilize other available resources to protect and/or to prevent further resource loss.

Consistent with the COR literature [16], we argue that psychological hardiness is one such protectively personal resource to buffer the negative effects of bullying on psychological stress. Indeed, psychological hardiness can aid stress resistance through its attributes of optimism and resiliency, which explains the tendency that hardy people appraise stressful events less threatening, thus diminishing stress symptoms [41]. As the three sub-dimensions of psychological hardiness were negatively associated with psychological stress [42], previous studies provided evidence for the contribution of psychological hardiness in buffering the negative impacts of stress on the well-being of employees. For instance, hardiness was found to moderate the effect of threats on psychological stress in a sample of 820 undergraduate students in the USA [46]. Workplace bullying can be a source of "threat" to employee work outcomes and well-being [47]. As a result, hardy employees can utilize their high hardiness to dampen the impacts of workplace bullying on their mental health [40, 48].

**H5a**: *Psychological hardiness moderates the relationship between workplace bullying and psychological stress such that higher psychological hardiness reduces the positive effect of workplace bullying on psychological bullying.*

There is support for psychological hardiness to dampen the influence of psychological stress on accidents and injuries among hardy workers. For instance, hardiness was found to buffer the effect of stress on illness [14]. Nurses with high hardiness in a high-stress circumstance tend to use 57% fewer sick hours than those reported low hardiness and low stress [39]. Hardy employees are likely to cope effectively with stressful work events, which lead to work performance and well-being. This will lead to a reduction in the likelihood of work-related accidents and injuries.

However, there were inconclusive findings on the relationship between psychological hardiness, stress, and health. There was empirical support for the moderation effect of hardiness on stress and symptoms of illness [49]. On the other hand, others did not find any empirical support for psychological hardiness as a moderator [11]. Therefore, more research is needed to further understand the buffering effect of hardiness on workplace accidents and injuries caused by work-related stress.

**H5b**: *Psychological hardiness moderates the relationship between psychological stress and workplace accidents and injuries such that higher psychological hardiness reduces the positive effect of psychological stress on workplace accidents and injuries.*

## Materials and methods

Written consent was granted from the ethics committee of Auckland University of Technology prior to data collection (AUT Ethics Committee; Written approval number: AUTEC Lamm 15/373). We conducted two studies to test the hypothesized model. We utilized IBM AMOS version 25 to conduct confirmatory factor analysis (CFA) for each of the previously validated scales used in this study. Cut-offs for the goodness of fit indices for the scale validity and the estimation of measurement model were consistent with the recommendation in the literature for structural equations modelling [50]. Written ethics approval was obtained from the university's ethics committee prior to data collection (AUT Ethics Committee; Written approval number: AUTEC Lamm 15/373).

### Overview of studies

We tested the hypotheses in two field studies. In Study 1, we tested the mediating effect of psychological stress on the relationship of workplace bullying with workplace accidents and injuries using a sample of Australian nurses. A second study was designed to replicate and validate the findings from Study 1. In addition, we test the buffering effects of psychological hardiness on workplace bullying → psychological stress → accidents and injuries (H5) with a sample of nurses from New Zealand.

### Study 1: Australian nurses

A market research company from Australia provided assistance with data collection. Australian nurses who were older than 18 years and resident of Australia were invited to complete the online survey in 2015 (approximate response rate 26%). We received completed surveys from 287 respondents. Most of the respondents (69.7%) were female and approximately half of the respondents were employed in public sector hospitals. Of the respondents, 85.7% were younger than 50 years old. Nearly two-thirds of the respondents did not hold a supervisory

position. Additionally, the majority (75.3%) were full-time nurses. Finally, most of the respondents had less than 5 years of organizational tenure.

## Measures

**Workplace bullying.** Workplace bullying was measured by using the nine-item NAQ-R short form [51]. Respondents were asked to indicate how often they experienced the negative acts at work, ranging from '1' never to '5' daily. A sample item included 'social exclusion from co-workers or work-group activities'.

**Psychological stress.** Psychological stress was measured using the K-10 Kessler Psychological Stress scale [52], ranging from '1' none of the time to '5' all of the time. A sample item was 'did you feel so nervous that nothing could calm you down?'

**Workplace accidents and injuries.** We used five types of workplace accidents injuries typically found in organizations [10, 11, 53, 54]. The respondents were asked to indicate their agreement on a 7-point Likert scale, ranging from 'very rarely/never' to 'very often (several times an hour). Sample items included 'Work-related accidents and or injuries from "trips, slips and falls"'.

**Control variables.** In this study, we controlled for: age [34], gender [14, 33], supervisory role [55], and organizational tenure [21]. These variables have been shown to have a confounding effect on the latent constructs.

## Discriminant analysis

To minimize common method variance (CMV), we utilized procedural remedies and post hoc statistical checks (such as Harman's single-factor model test and "social desirability" scale as the marker variable) to ensure CMV is of no major concern [56]. Harman's single-factor resulted in five factors where the largest factor accounted for 39.2% variance. The Marlowe-Crowne Social Desirability scale [57] was incorporated into the model as marker variable [58]. The test showed that the difference of the correlations between exogenous and endogenous variables before and after including the marker variable was 0.09, below the cut-off value of 0.2 [58]. These findings indicated that CMV was not a major issue in this study.

We also conducted Chi-square nested model tests to compare the changes in $\lambda^2$ of the hypothesized three-factor model with that of alternative models for discriminant validity (Table 1). Results showed that the preferred model has a satisfactory fit with the data ($\lambda^2$/df = 1.923, df = 224, CFI = 0.975, TLI = 0.969, RMSEA = 0.057, SRMR = 0.037).

## Results: Study 1

Descriptive statistics and intercorrelations are reported in Table 2. Male nurses experienced higher level of workplace bullying ($\beta$ = 0.18, $p$<0.01) and more workplace accidents and

**Table 1. Results for Chi-square difference test in Study 1.**

| Model | $\chi^2$ | df | CFI | TLI | RMSEA | SRMR | $\Delta\chi^2$ from 3-factor model |
|---|---|---|---|---|---|---|---|
| 3-factor model (WB, PS, A&I) | 430.72 | 224 | 0.975 | 0.969 | 0.057 | 0.0371 | |
| 2-factor model (WB, PS+A&I) | 1137.709 | 226 | 0.889 | 0.864 | 0.119 | 0.1108 | 706.989 (df = 2), *** |
| 1-factor model (WB+PS+A&I) | 1993.598 | 227 | 0.785 | 0.738 | 0.165 | 0.1290 | 885.889 (df = 1), *** |

*Note*: WB = workplace bullying; PS = psychological stress; A&I = accidents and injuries

\*\*\*$p$<0.001.

**Table 2. Descriptive statistics and inter-correlations in Study 1.**

|  | Mean | SD | 1 | 2 | 3 | 4 | 5 | 6 | 7 |
|---|---|---|---|---|---|---|---|---|---|
| 1. Age | 1.70 | 0.46 | 1.00 |  |  |  |  |  |  |
| 2. Gender | 3.99 | 1.30 | 0.05 | 1.00 |  |  |  |  |  |
| 3. Supervisory role | 1.61 | 0.49 | 0.10 | -0.06 | 1.00 |  |  |  |  |
| 4. Tenure | 3.02 | 1.22 | 0.01 | 0.46** | -0.07 | 1.00 |  |  |  |
| 5. WB | 1.93 | 0.94 | -0.18** | -0.30*** | -0.10 | -0.21*** | 1.00 |  |  |
| 6. PS | 2.30 | 0.98 | -0.07 | -0.34*** | 0.00 | -0.33*** | 0.71*** | 1.00 |  |
| 7. A&I | 3.32 | 1.61 | -0.22*** | -0.27*** | -0.11 | -0.28*** | 0.57*** | 0.56*** | 1.00 |

*Note*: N = 287 Australian nurses, Gender ('1' male, '2' female), Supervisory role ('1' yes, '0' no), SD: standard deviation

**p<0.01

***p<0.001.

injuries ($\beta = 0.22$, $p<0.001$). Older respondents had more tenure in their hospital ($\beta = 0.46$, $p<0.01$). These control variables were incorporated into model testing.

Path analysis showed that workplace bullying had a positive association with psychological stress ($\beta = 0.72$, $p<0.001$) and workplace accidents and injuries ($\beta = 0.52$, $p<0.001$). Psychological stress had a positive association with workplace accidents and injuries ($\beta = 0.29$, $p<0.001$). Psychological stress was a partial mediator of the effect of workplace bullying on accidents and injuries (effect = 0.351, se = 0.089, 95%CI = 0.219, 0.510).

In summary, Study 1 provided empirical support for Hypotheses 1 to 4. Consistent with the literature and the AET perspective [23], nurses reacted negatively to having experienced negative workplace events such as bullying exposure results in psychological stress [5]. We contributed to the literature by empirically showed the direct relationship between psychological stress and workplace accidents and injuries.

## Study 2: New Zealand nurses

Study 2 was designed to test the moderated mediation model. Data were collected from a cross-sectional sample of 201 New Zealand nurses in 2016. Most of the respondents were female and the majority were from the North Island (such as Auckland, Wellington, and Hamilton) and Christchurch in the South Island. Nearly half of the respondents were between 26–40, followed by 51–60 years old (15.4%). A large majority were full-time employees.

### Measures

We used the same three variables from Study 1 with the same rating scales. Similar to Study 1, we controlled for confounding effects with the same set of demographic variables. Also, we introduced psychological hardiness as a moderator into Study 2 to replicate the findings from Study 1 and to test the moderation hypotheses. Psychological hardiness was measured using a six-item scales [15], ranging from '1' strongly disagree to '7' strongly agree. We used a total score approach by combining the scores from the three sub-dimensions (commitment, control, and challenge) into a second order, composite hardiness score (sample item includes 'despite setbacks, I remain committed to accomplishing job tasks').

### Discriminant analysis

Discriminant analysis was undertaken with a series of Chi-square nested model tests (see Table 3), by comparing the four-factor (hypothesized model) with alternate models. Results

**Table 3. Results for Chi-squared comparison test in Study 2.**

| Model | $\chi^2$ | df | CFI | TLI | RMSEA | SRMR | $\Delta\chi^2$ from 4-factor model |
|---|---|---|---|---|---|---|---|
| 4-factor model (WB, PS, PH, A&I) | 800.074 | 463 | 0.938 | 0.929 | 0.060 | 0.063 | |
| 3-factor model (WB, PS, PH+A&I) | 1180.462 | 466 | 0.868 | 0.851 | 0.088 | 0.107 | 380.388 (df = 3)*** |
| 2-factor model (WB, PS+PH+A&I) | 1566.952 | 468 | 0.798 | 0.772 | 0.108 | 0.124 | 386.49 (df = 2)*** |
| 1-factor model (WB+PS+PH+A&I) | 1808.407 | 469 | 0.753 | 0.722 | 0.119 | 0.138 | 241.455 (df = 1)*** |

*Note*: WB = workplace bullying; PS = psychological stress; PH = psychological hardiness; A&I = accidents and injuries

***$p < 0.001$.

showed that the hypothesized four-factor model had the best fit ($\chi^2$/df = 1.728, CFI = 0.94, TLI = 0.93, RMSEA = 0.06, SRMR = 0.07). We conducted the incorporation of a marker variable (social desirability) to check for CMV. The test showed that the difference of the correlations between endogenous and exogenous variables before and after including the marker variable was 0.13, below the cut-off value of 0.2 [58], indicating that CMV was not a major issue.

## Results

Descriptive statistics and intercorrelation coefficients are reported in Table 4. Female respondents were older, held supervisory positions, and had more tenure. Supervisors had more experienced in the organization. As a result of the bivariate analysis, these control variables were also incorporate into the path analyses.

Results of the path modelling showed tenure to have a negative association with psychological stress ($\beta$ = -0.13, $p < 0.05$) and workplace accidents and injuries ($\beta$ = -0.21, $p < 0.001$), respectively. As shown in Fig 2, there was support for Hypotheses 1–4 which demonstrated a mediational model (b = 0.15, SE = 0.05, 95%CIs = 0.07, 0.24). Using the Johnson–Neymann (J–N) technique, we found psychological hardiness was found to buffer the effect of workplace bullying on workplace accidents and injuries (b = -0.20, SE = 0.04, 95%CIs = -0.25, -0.49, $p < 0.001$). Mediation analysis was performed using the estimand plug-in [59] in IBM AMOS version 25. Psychological hardiness was found to be a moderator of the indirect effect of workplace bullying on workplace accidents and injuries as mediated by psychological stress (b = 0.12, SE = 0.05, 95%CIs = 0.04, 0.22, $p < 0.05$). This finding provides support for Hypothesis 5b. As shown by the moderation plot (Fig 3), in the presence of high psychological

**Table 4. Descriptive statistics and inter-correlations in Study 2.**

| | Mean | SD | 1 | 2 | 3 | 4 | 5 | 6 | 7 | 8 |
|---|---|---|---|---|---|---|---|---|---|---|
| 1. Age | 3.15 | 1.55 | 1.00 | | | | | | | |
| 2. Gender | 1.76 | 0.43 | 0.23** | 1.00 | | | | | | |
| 3. SR | 0.30 | 0.46 | 0.30*** | 0.07 | 1.00 | | | | | |
| 4. Tenure | 2.99 | 1.31 | 0.57*** | 0.23** | 0.28*** | 1.00 | | | | |
| 5. WB | 1.73 | 0.85 | -0.22** | -0.22** | 0.03 | -0.22** | 1.00 | | | |
| 6. PS | 1.06 | 0.39 | -0.30*** | -0.13 | -0.08 | -0.33*** | 0.60*** | 1.00 | | |
| 7. PH | 4.42 | 0.64 | 0.24*** | 0.30*** | 0.14 | 0.23** | -0.30*** | -0.31*** | 1.00 | |
| 8. A&I | 1.49 | 0.90 | -0.32*** | -0.25*** | -0.02 | -0.30*** | 0.66*** | 0.59*** | -0.38*** | 1.00 |

*Note*: N = 201 New Zealand nurses, Gender ('1' male, '2' female), SR: Supervisory role ('1' yes, '0' no), SD: Standard Deviation

**$p < 0.01$

***$p < 0.001$.

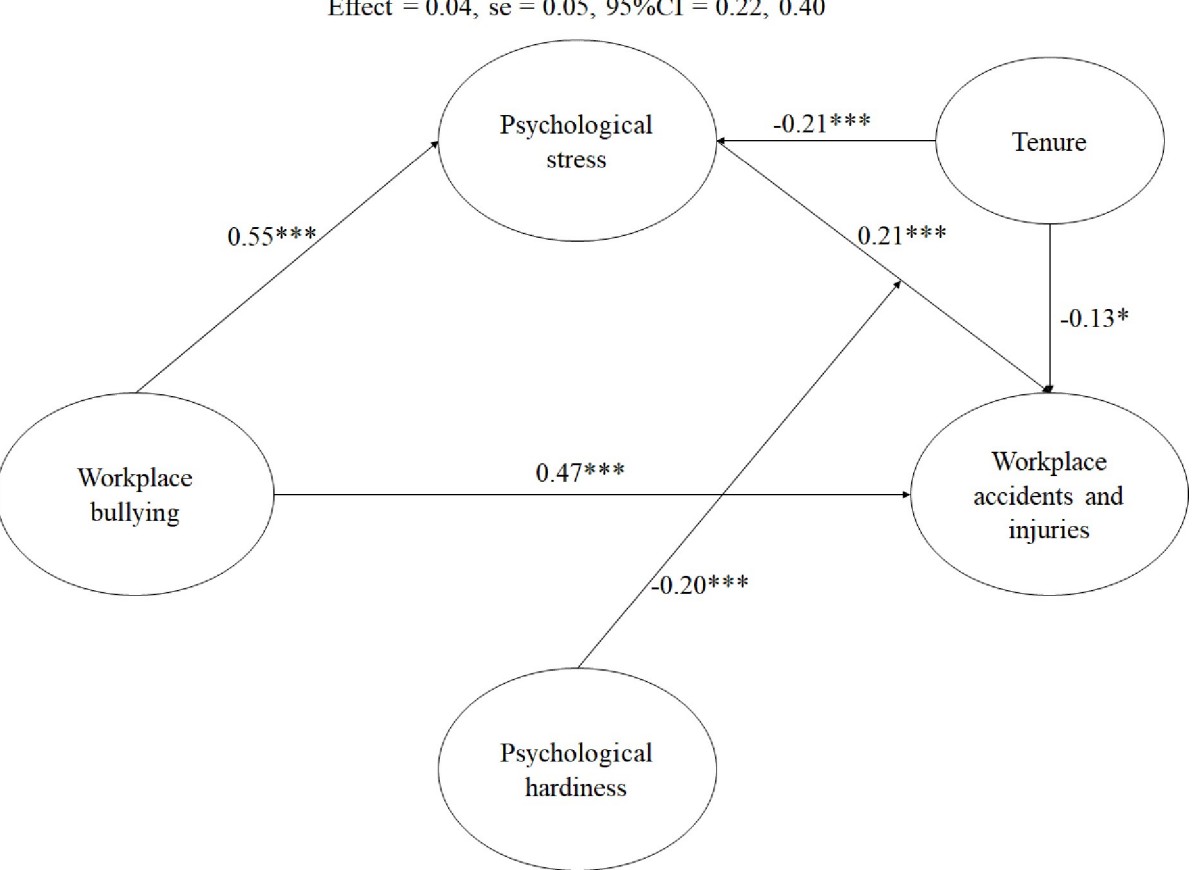

Fig 2. Results of hypotheses testing in Study 2. *Note*: N = 201, *p<0.05, ***p<0.001.

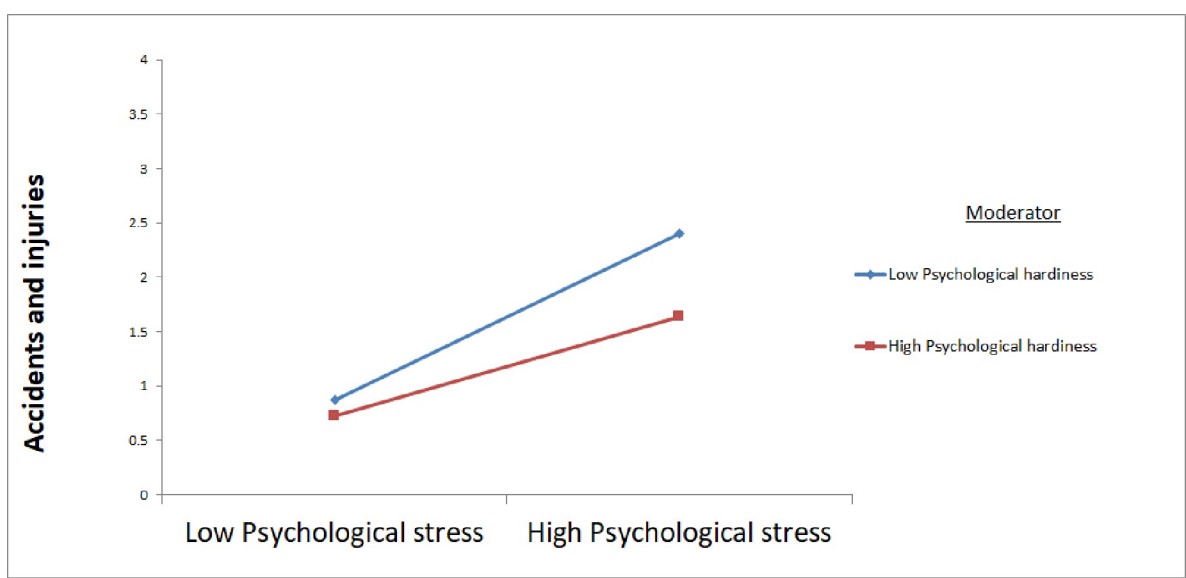

Fig 3. Moderation of psychological hardiness on stress and accidents and injuries.

hardiness, this would reduce the negative impact of psychological stress on workplace accidents and injuries. In summary, the model supported a moderated mediation model where psychological stress was a partial mediator of the effect of workplace bullying on workplace accidents and injuries.

## Discussion

### Theoretical implications

This study aimed to test a moderated mediation model using two samples of nurses where we hypothesized that the negative impact of workplace bullying on accidents and injuries at work would be mediated by psychological stress. The indirect effect of workplace bullying on workplace accidents and injuries through psychological stress were then moderated by the degree of psychological hardiness. Our findings contribute new insights into the limited knowledge of the workplace safety of nurses in the context of workplace bullying.

The AET [23] posits that employees would react emotionally to the negative affective events like workplace bullying that would affect their subsequent behaviors, attitudes, and well-being in responding to the respective event [5]. We supported previous studies [45] by providing strong support for a partial mediation model, where psychological stress as the mediator, contributed to workplace accidents and injuries when nurses encounter workplace bullying. Stated differently, our study provided support for the evidence that workplace bullying exposure could result in accidents and injuries [34] via psychological stress [5].

As the COR theory [16] posits, the prevalence of workplace bullying is perceived to be a stressful working condition that could develop perceptions of substantial resource threats or loss or the experience of resource depletion [45]. As previously argued, psychological hardiness is a type of protective personal resources that individuals can utilize to preserve other resources or prevent further resource loss in coping with stress [38, 43]. While we did not find the role of psychological hardiness in buffering the negative consequences of workplace bullying on psychological stress (H5a), we did find psychological hardiness to buffer the effect of psychological stress on workplace accidents and injuries (H5b). This was due to the partial mediation effect of psychological stress caused by workplace bullying (H4). The significant finding in our study supported the existing literature using psychological hardiness as the moderator for psychological stress [17, 46]. Indeed, individuals with high positive affectivity are less likely to perceive workplace bullying and are of low risk to become targets of bullying [60]. On the other hand, individuals high in negative affectivity are more likely to feel bullied or mistreated because they are more sensitive and more reactive to negative events [61]. Therefore, we concluded that nurses who are high in hardiness are less likely to experience bullying than those who are low in hardiness.

### Practical and managerial implications

Our findings have several practical implications. To minimize the prevalence of workplace bullying, several strategies could be used to attract and select managers who do not demonstrate laissez-faire leadership behaviors [62] or Machiavellian behaviors [63]. Similarly, selection strategies could be adopted to ensure newcomers do not fit the profile of bullies [30]. Senior management must take an ethical stand in promoting "ethical work environments devoid of interpersonal mistreatment" [64], which is consistent with the work environment hypothesis in explaining workplace bullying [7].

Various strategies could be developed to build on the three dimensions of psychological hardiness (such as commitment, control, and challenge). For example, HR managers could consider the implementation of educational programs to improve the resilience [8] and or

psychological hardiness of nurses [65]. Henderson used a hardiness training program as an intervention tool to educate nurses on developing strategies to strengthen the three dimensions of psychological hardiness (i.e. commitment, control, and challenge) and supplemented this with specific strategies for nurses to practice their assertiveness, active involvement (instead of avoidance) in stressful events and view these challenges as growth opportunity [65]. Through education intervention programs, nurses would be given the psychological resources to buffer the negative consequences of negative workplace behaviors [10]. This is consistent with the COR perspective [16].

The above intervention examples were included in a recently developed typology [66]. Caponecchia et al. identified 11 core intervention types (investigation, codes of conduct, policy; EAP and counselling, bullying awareness training, coaching, system-wide intervention, skills training and development, values statements, local resolution, and organizational redesign). HR professionals could consider implementing these interventions to enhance the well-being of nurses. More research is also needed to examine the effectiveness of primary versus secondary versus tertiary interventions [67] as these could have a different impact on training nurses to effectively deal with their exposure to workplace bullying.

## Limitations and future implications

The current study found that CMV is not a major concern through some statistical checks for common method variance as well as the significant impacts of moderated mediation [56]. We still acknowledged that the findings could potentially be affected by a single source bias. Therefore, future studies could rely on objective organizational data on accidents and injuries instead of relying on self-reported data from the participants.

Consistent with the AET, future research could adopt self-regulation theory to collect multi-source data to evaluate how nurses regulate their emotions resulting from the exposure to workplace bullying [68]. Another possibility is to design studies to collect longitudinal data to test for the effect of workplace bullying on individual and work safety outcomes [24]. Future study could potentially collect data such as the ward/unit as these could control for unit-level variances on work environment, which could affect the prevalence of exposure to workplace bullying [69, 70].

## Conclusion

In conclusion, psychological hardiness was found to be a moderator of the partial mediation effect of psychological stress due to workplace bullying on workplace accidents and injuries. This study contributed to the knowledge of psychological hardiness and was consistent with literature [15]. We concluded that when nurses possessed a high level of psychological hardiness, they were better at being resilient and possess the ability to cope effectively when they experience negative treatments at work.

## Supporting information

**S1 File.**
(DOC)

## Author Contributions

**Conceptualization:** Stephen T. T. Teo, Diep Nguyen, Fiona Trevelyan.

**Data curation:** Stephen T. T. Teo.

**Formal analysis:** Stephen T. T. Teo, Diep Nguyen.

**Funding acquisition:** Stephen T. T. Teo, Felicity Lamm, Mark Boocock.

**Investigation:** Stephen T. T. Teo.

**Methodology:** Stephen T. T. Teo, Diep Nguyen.

**Project administration:** Stephen T. T. Teo.

**Resources:** Fiona Trevelyan, Felicity Lamm.

**Software:** Diep Nguyen.

**Writing – original draft:** Stephen T. T. Teo, Diep Nguyen.

**Writing – review & editing:** Fiona Trevelyan, Felicity Lamm, Mark Boocock.

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
