## [Decision Letter · Decision Letter 0]

27 Oct 2020

PONE-D-20-12891

Workplace Bullying, Psychological Hardiness, and Accidents and Injuries in Nursing: A Moderated Mediation Model

PLOS ONE

Dear Dr. Teo,

Thank you for submitting your manuscript to PLOS ONE. After careful consideration, we feel that it has merit but does not fully meet PLOS ONE’s publication criteria as it currently stands. Therefore, we invite you to submit a revised version of the manuscript that addresses the points raised during the review process.

Your paper has been reviewed by two acknowledged experts in the field of knowledge covered by the study. Overall, the comments are very positive, and the paper seems to have good potential to be considered for publication in PLOS ONE. Nevertheless, there are some comments and queries provided by the reviewers that need your attention, through a revision of the manuscript. Concretely, some miscellaneous (but essential) points need to be clarified, focusing in the study setting and the ethical proceeding followed for its performance. Additionally, the limitations of the study must be improved and explicitly stated and discussed in the light of their potential impact on the results. Finally, the second Reviewer states that the manuscript is excessively long; please try to synthesize some parts of the paper, where the information is repetitive or redundant.

We look forward to receiving your revised manuscript.

Kind regards,

Sergio A. Useche, Ph.D.

Academic Editor

PLOS ONE

Journal Requirements:

'The funders had no role in study design, data collection and analysis, decision to publish, or preparation of the manuscript.'

Reviewers' comments:

Reviewer's Responses to Questions

**Comments to the Author**

1. Is the manuscript technically sound, and do the data support the conclusions?

Reviewer #1: Yes

Reviewer #2: Yes

2. Has the statistical analysis been performed appropriately and rigorously? 

Reviewer #1: Yes

Reviewer #2: Yes

3. Have the authors made all data underlying the findings in their manuscript fully available?

Reviewer #1: Yes

Reviewer #2: Yes

4. Is the manuscript presented in an intelligible fashion and written in standard English?

Reviewer #1: Yes

Reviewer #2: Yes

5. Review Comments to the Author

Reviewer #1: It is a very well written manuscript. You have studied a really important field and you reached some valuable conclusions that reseach community can gain from that. Methodology and results are appropriatelly described.

Reviewer #2: dear Authors

My suggestions as bellow:

1- Please update reference Fevre and colleagues (2011) and information about the UK.

2- Please write research ethics: ethical code, getting Satisfaction.

3- Please add time of research.

4- Please describe research limitations.

5- I think type of wards in which the nurses studied worked is important. Can be added.

6- Manuscript is too long.

6. PLOS authors have the option to publish the peer review history of their article (what does this mean?). If published, this will include your full peer review and any attached files.

Reviewer #1: No

Reviewer #2: No

---

## [Author Response · Author response to Decision Letter 0]

6 Nov 2020

06 Nov 2020 Comment from the Journal Editorial Office:

1.) Please confirm whether the data now located in your manuscript and/or Supporting information files constitutes the minimal data set. The minimal data set is defined as the data set used to reach the conclusions drawn in the manuscript with related metadata and methods, and any additional data required to replicate the reported study findings in their entirety. This may include: a.) The values behind the means, standard deviations and other measures reported; b.) The values used to build graphs; c.) The points extracted from images for analysis (https://journals.plos.org/plosone/s/data-availability#loc-minimal-data-set-definition).

I can confirm this requirement has been addressed.

2.) If so, please confirm whether the following information is accurate, and whether we may update your Data Availability Statement accordingly on your behalf:

"The full data set cannot be shared publicly because formal approval was not granted by the Ethics Committee. However, all relevant data necessary to replicate the study's results are within the manuscript and its Supporting Information files."

The above statement is correct.

We have also addressed all of the additional requirements requested by the Editorial Office - please refer to our letter.

Reviewer 1: It is a very well written manuscript. You have studied a really important field and you reached some valuable conclusions that research community can gain from that. Methodology and results are appropriately described.

Authors: Thank you for your support and comments about the contribution of our paper.

Reviewer 2:

1. Please update reference Fevre and colleagues (2011) and information about the UK.

Authors: We have updated the reference according to this journal’s preferred referencing style. 

Fevre et al (2011) and the UK context was correctly referenced in the original manuscript (please refer to page 3). We can confirm that the original reference used in the manuscript was correct (see https://www.researchgate.net/publication/258206731_Insight_into_ill-treatment_patterns_causes_and_solutions/link/57fb640608ae8da3ce60d58d/download). 

2. Please write research ethics: ethical code, getting Satisfaction.

Authors: This was reported in the manuscript portal: Auckland University of Technology Ethics Committee (number: Lamm 15373). 

3. Please add time of research.

Authors: The research was undertaken over 2015 and 2016 as there were two studies in the paper. Please refer to p. 10 and p.12.

4. Please describe research limitations.

Authors: Research limitation was already written into the original manuscript. Please see the revised section on page 16-17. 

5. I think type of wards in which the nurses studied worked is important. Can be added.

Authors: This is a great idea. As the study is based on a cross sectional dataset, we did not collect any information on the ward of the participants. This is now listed as a suggestion for future study (see page 16-17).

6. Manuscript is too long.

Authors: We have shortened the manuscript and the number of references. Its current length is 4670 words (excluding references, tables, and figures).

---

## [Decision Letter · Decision Letter 1]

10 Dec 2020

Workplace Bullying, Psychological Hardiness, and Accidents and Injuries in Nursing: A Moderated Mediation Model

PONE-D-20-12891R1

Dear Dr. Teo,

We’re pleased to inform you that your manuscript has been judged scientifically suitable for publication and will be formally accepted for publication once it meets all outstanding technical requirements.

Kind regards,

Sergio A. Useche, Ph.D.

Academic Editor

PLOS ONE

Additional Editor Comments (optional): Authors have done a good revision. The paper can be accepted as it is.

Reviewers' comments:

Reviewer's Responses to Questions

**Comments to the Author**

1. If the authors have adequately addressed your comments raised in a previous round of review and you feel that this manuscript is now acceptable for publication, you may indicate that here to bypass the “Comments to the Author” section, enter your conflict of interest statement in the “Confidential to Editor” section, and submit your "Accept" recommendation.

Reviewer #1: All comments have been addressed

Reviewer #2: All comments have been addressed

2. Is the manuscript technically sound, and do the data support the conclusions?

Reviewer #1: Yes

Reviewer #2: Yes

3. Has the statistical analysis been performed appropriately and rigorously? 

Reviewer #1: Yes

Reviewer #2: N/A

4. Have the authors made all data underlying the findings in their manuscript fully available?

Reviewer #1: Yes

Reviewer #2: (No Response)

5. Is the manuscript presented in an intelligible fashion and written in standard English?

Reviewer #1: Yes

Reviewer #2: Yes

6. Review Comments to the Author

Reviewer #1: (No Response)

Reviewer #2: (No Response)

7. PLOS authors have the option to publish the peer review history of their article (what does this mean?). If published, this will include your full peer review and any attached files.

Reviewer #1: No

Reviewer #2: No

---

## [Editor Report · Acceptance letter]

15 Dec 2020

PONE-D-20-12891R1 

Workplace Bullying, Psychological Hardiness, and Accidents and Injuries in Nursing:
A Moderated Mediation Model 

Dear Dr. Teo:

I'm pleased to inform you that your manuscript has been deemed suitable for publication in PLOS ONE. Congratulations! Your manuscript is now with our production department. 

Kind regards, 

on behalf of

Dr. Sergio A. Useche 

Academic Editor

PLOS ONE